# Parametric Instance Classification for Unsupervised Visual Feature Learning

**Yue Cao**[*1], **Zhenda Xie**[*12], **Bin Liu**[*12], **Yutong Lin**[13], **Zheng Zhang**[1], **Han Hu**[1]
[1]Microsoft Research Asia
[2]Tsinghua University
[3]Xi'an Jiaotong University
{yuecao,t-zhxie,v-liubin,v-yutlin,zhez,hanhu}@microsoft.com

## Abstract

This paper presents parametric instance classification (PIC) for unsupervised visual feature learning. Unlike the state-of-the-art approaches which do instance discrimination in a dual-branch non-parametric fashion, PIC directly performs a one-branch parametric instance classification, revealing a simple framework similar to supervised classification and without the need to address the information leakage issue. We show that the simple PIC framework can be as effective as the state-of-the-art approaches, i.e. SimCLR and MoCo v2, by adapting several common component settings used in the state-of-the-art approaches. We also propose two novel techniques to further improve effectiveness and practicality of PIC: 1) a sliding-window data scheduler, instead of the previous epoch-based data scheduler, which addresses the extremely infrequent instance visiting issue in PIC and improves the effectiveness; 2) a negative sampling and weight update correction approach to reduce the training time and GPU memory consumption, which also enables application of PIC to almost unlimited training images. We hope that the PIC framework can serve as a simple baseline to facilitate future study. The code and network configurations are available at `https://github.com/bl0/PIC`.

## 1 Introduction

Visual feature learning has long been dominated by supervised image classification tasks, e.g. ImageNet-1K classification. Recently, unsupervised visual feature learning has started to demonstrate on par or superior transfer performance on several downstream tasks compared to supervised approaches [15, 5, 22]. This is encouraging, as unsupervised visual feature learning could utilize nearly unlimited data without annotations.

These unsupervised approaches [15, 22, 5, 6], which achieve revolutionary performance, are all built on the same pre-text task of instance discrimination [12, 32, 27], where each image instance is treated as a distinct class. To solve the instance discrimination task, often a dual-branch structure is employed where two augmentation views from the same image are encouraged to agree and views from different images to disperse. In general, for dual-branch approaches, special designs are usually required to address the information leakage issue, e.g. specialized networks [1], specialized BatchNorm layers [15, 5], momentum encoder [15], and limited negative pairs [5].

Unlike these dual-branch non-parametric approaches, this paper presents a framework which solves instance discrimination by direct parametric instance classification (PIC) [12]. PIC is a one-branch scheme where only one view for each image is required per iteration, which avoids the need to

carefully address the information leakage issue. PIC can also be easily adapted from the simple supervised classification frameworks. We additionally show that PIC can be as effective as the state-of-the-art approaches [6, 5] by adopting several recent advances, including a cosine soft-max loss, a stronger data augmentation and a 2-layer projection head.

There remain issues regarding effectiveness and practicality on large data. Specifically, PIC faces an extremely infrequent instance visiting issue, where each instance is visited as positive only once per epoch, which hinders representation learning in the PIC framework. PIC also has a practicality issue with respect to training time and GPU memory consumption, as there is a large classification weight matrix to use and update.

Two novel techniques are proposed to address these two issues. The first is a sliding window based data scheduler to replace the typical epoch-based one, to shorten the distance between two visits of the same instance class for the majority of instances. It proves to significantly speed up convergence and improve the effectiveness. The second is a negative instance class sampling approach for loss computation along with weight update correction, to make the training time and GPU memory consumption near constant with increasing data size while maintaining effectiveness.

We hope the PIC framework could serve as a simple baseline to facilitate future study, because of its simplicity, effectiveness, and ability to incorporate improvements without considering the information leakage issue. The code and network configurations are available at `https://github.com/bl0/PIC`.

## 2 Methodology

### 2.1 Parametric Instance Classification (PIC) Framework

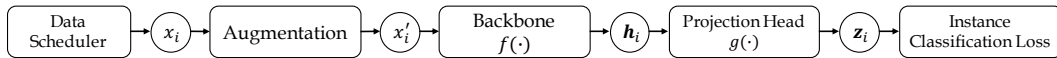

**Figure 1:** An illustration of the PIC framework.

PIC learns representations by parametric instance classification, where each instance is treated as a distinct class. Like common supervised classification frameworks [16], it consists of five major components:

  i. a *data scheduler* that feeds training images into networks during the course of training;

  ii. a *data augmentation module* that randomly augments each training example fed into the network;

  iii. a *backbone network* that extracts a feature map for each augmented image, which will also be transferred to downstream tasks;

  iv. a *small projection head* that projects the feature map to a feature vector for which the instance classification loss is applied;

  v. an *instance classification loss* that penalizes the classification errors in training.

While directly applying the usual component settings of supervised category classification for PIC will result in poor transfer performance as shown in Table 1, we show that there is no intrinsic limitation in the PIC framework, in contrast to the inherent belief in previous works [32]. We show that poor performance is mainly due to improper component settings. By replacing several usual component settings with the ones used in recent unsupervised frameworks [5, 6], including *a cosine soft-max loss*, *a stronger data augmentation* and *a 2-layer MLP projection head*, the transfer performance of the learnt features in the PIC framework are significantly improved. The *cosine soft-max loss* is commonly used in metric learning approaches [29, 30] and also in recent state-of-the-art unsupervised learning frameworks [15, 5], as

$$L = -\frac{1}{|B|} \sum_{i \in B} \log \frac{\exp\left(\cos\left(\mathbf{w}_i, \mathbf{z}_i\right)/\tau\right)}{\sum_{j=1}^{N} \exp\left(\cos\left(\mathbf{w}_j, \mathbf{z}_j\right)/\tau\right)}, \tag{1}$$

where $\mathbf{W} = [\mathbf{w}_1, \mathbf{w}_2, ..., \mathbf{w}_N] \in \mathbb{R}^{D \times N}$ is the parametric weight matrix of the cosine classifier; $B$ denotes the set of instance indices in a mini-batch; $\mathbf{z}_i$ is the projected feature for instance $i$;

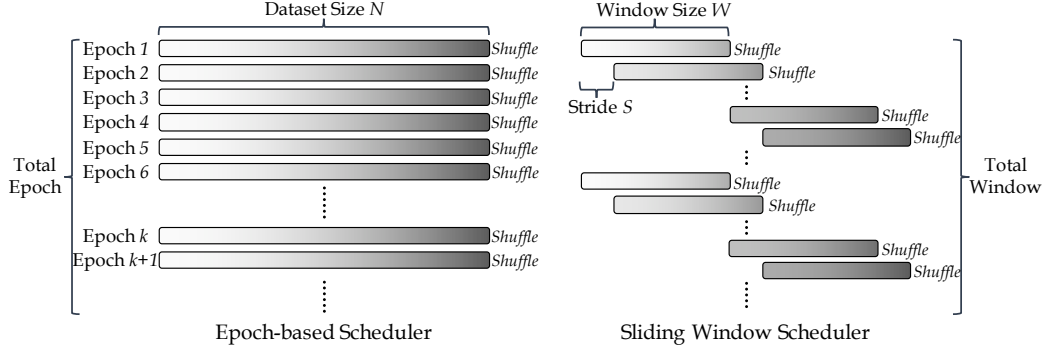

**Figure 2:** An illustration of the sliding window data scheduler.

$\cos(\mathbf{w}_j, \mathbf{z}_i) = (\mathbf{w}_j \cdot \mathbf{z}_i)/(\|\mathbf{w}_j\|_2 \cdot \|\mathbf{z}_i\|_2)$ is the cosine similarity between $\mathbf{w}_j$ and $\mathbf{z}_i$; and $\tau$ is a scalar temperature hyper-parameter. While the choice of standard or cosine soft-max loss in supervised classification is insignificant, the cosine soft-max loss in PIC performs significantly better than the standard soft-max loss. The *stronger data augmentation* and *2-layer MLP projection head* are introduced in [5], and we find they also benefit the PIC framework.

There are still obstacles preventing the PIC framework from better representation learning, e.g. the optimization issue caused by too infrequent visiting of each instance class (see Section 2.2). The PIC framework also has practicality problems regarding training time and GPU memory consumption, especially when the data size is large (see Section 2.3). To further improve the quality of learnt feature representations as well as to increase practicality, two novel techniques are proposed:

(a) A novel *sliding window data scheduler* to replace the usual epoch-based data scheduler. The new scheduler well addresses the issue that each instance class is visited too infrequently in unsupervised instance classification (e.g. once per epoch).

(b) A negative instance class sampling approach for loss computation along with *weight update correction*, which makes the training time and GPU memory consumption near constant with increasing data size, and maintains the effectiveness of using all negative classes for loss computation.

With the above improvements, the PIC framework can be as effective as the state-of-the-art frameworks, such as SimCLR [5] and MoCo v2 [6]. It is also practical regarding training time and GPU memory consumption.

## 2.2 Sliding Window Data Scheduler

When a regular epoch-based data loader is employed, each *instance* class will be visited exactly once per epoch, which is extremely infrequent when the training data is large, e.g. more than 1 million images. The extremely infrequent class visits may affect optimization and likely result in sub-optimal feature learning. While it may benefit optimization to decrease the visiting interval of each instance class, it is theoretically difficult in the sense of expectation, according to the following proposition.

**Proposition 1**. Denote the number of images in an epoch as $N$. For an arbitrary data scheduler which visits each instance once per epoch, the expectation of distance between two consecutive visits of the same instance class is $N$.

To address this dilemma, our idea is to maintain a relatively low distance between visits (denoted as $D$) for the *majority* (the majority ratio is denoted as $\gamma$) of instance classes during training. To this goal, we propose a sliding window data scheduler, as illustrated in Figure 2. In contrast to the regular data scheduler which traverses over the whole set of training images epoch-by-epoch, the proposed sliding window data scheduler traverses images within windows, with the next window shifted from the previous one. There are overlaps between successive windows, and the overlapped instance classes are regarded as the *majority*, and thus visited twice in a relatively short time.

Hence, the window size $W$ is equal to the distance between consecutive visits for majority classes, $W = D$. And the sliding stride $S$ is the number of instances by which the window shifts for each step, $S = (1 - \gamma) \times W = (1 - \gamma) \times D$, where $\gamma$ represents the majority ratio within a window. As

the majority ratio $\gamma$ and the average distance $D$ between consecutive visits for majority classes are both constant for increasing number of total training images, this indicates that larger training data leads to almost no increase of the optimization issue by using the sliding window data scheduler. Like the regular data scheduler, before each traversal, images in the current window are shuffled to introduce stochastic regularization.

In experiments, for the majority ratio $\gamma = 87.5\%$ with relatively low visiting distance $D = 2^{17} = 131,072$, we need to set the window size to $W = 131,072$ and the sliding stride to $S = 2^{14} = 16,384$. This performs noticeably better than the regular data scheduler on the ImageNet-1k dataset, which has 1.2M training images. For different data scales, the window size $W$ and stride $S$ may be adjusted accordingly to obtain better performance.

## 2.3 Negative Instance Sampling and Classification Weight Correction

The vanilla PIC framework has a practical limitation on large-scale data (e.g billions of images) during the training phase. This limitation comes from two aspects: 1) in the forward/backward phase, logits of all negative classes are employed in the denominator of the soft-max computation in Eq. (1); 2) in the weight update phase, since the commonly used SGD typically has *weight decay* and *momentum* terms, even if weights in the instance classifier are not used in the current batch, they will still be updated and synchronized. As a result, the training time and GPU memory consumption are linearly increased w.r.t the data size, limiting practicability on large-scale data.

We propose two approaches to significantly reduce the training time and GPU memory consumption, making them near constant with increasing data size. The first is *recent negative sampling* to address the issues in the forward/backward phase, and the second is *classification weight update correction* to address the issues in the weight update phase.

**Recent negative sampling** employs only the most recent $K$ instances in recent iterations as the negative instance classes in the soft-max computation, which appear in the denominator of the cosine soft-max loss in Eq. (1). Hence, the computation costs in the forward phase to compute the loss and the backward phase to compute the gradients are reduced. As shown in Table 2, we find that $K = 65536$ achieves accuracy similar to the counterpart using all instances (about 1.28M) with 200-epoch pre-training on ImageNet.

**Classification weight update correction** In the SGD optimizer, the weights are updated as:

$$\mathbf{u}_i^{(t+1)} := m\mathbf{u}_i^{(t)} + (\mathbf{g}_i^{(t)} + \lambda\mathbf{w}_i^{(t)}), \quad \mathbf{w}_i^{(t+1)} := \mathbf{w}_i^{(t)} - \eta\mathbf{u}_i^{(t+1)}, \tag{2}$$

where $\mathbf{g}_i^{(t)}$ and $\mathbf{u}_i^{(t)}$ are the gradient and momentum terms at iteration $t$ for classification weight $\mathbf{w}_i^{(t)}$ of instance class $i$, respectively; $\lambda$, $m$ and $\eta$ are the weight decay, momentum scalar and learning rate, respectively. Even when a negative instance class $i$ is not included in the denominator of the soft-max loss computation, where $\mathbf{g}_i^{(t)} = \mathbf{0}$, the classification weight vector $\mathbf{w}_i^{(t)}$ will still be updated due to the *weight decay* and *momentum* terms being non-zero, as shown in Eq. (2). Hence, GPU memory consumption in the weight update and synchronization process cannot be reduced. If we directly ignore the weight decay and momentum terms for the non-sampled negative classes, the different optimization statistics between sampled and non-sampled negative classes lead to a significant decrease in accuracy.

Noticing that the update trajectories for the classification weights of non-sampled negative classes are predictable (only affected by the weight decay and momentum terms), we propose a closed-form one-step correction approach to account for the effect of momentum and weight decay:

$$\begin{pmatrix} \mathbf{w}_i^{(t+t')} \\ \mathbf{u}_i^{(t+t')} \end{pmatrix} := \begin{pmatrix} 1 - \eta \cdot \lambda & -\eta \cdot m \\ \lambda & m \end{pmatrix}^{t'} \begin{pmatrix} \mathbf{w}_i^{(t)} \\ \mathbf{u}_i^{(t)} \end{pmatrix} \tag{3}$$

where $t'$ is the distance from the last visit to the current visit for instance class $i$. The detailed deduction and implementation are shown in Appendix B.

Hence, the non-sampled negative classes are not required to be stored in GPU memory and updated at the current iteration, and the weight correction procedure is performed the next time these classes are sampled. The GPU memory consumption is thus determined by the number of sampled negatives $K$, and are constant with respect to the data size.

# 3  Related Works

**Pre-text tasks**  Unsupervised visual feature learning is usually centered on selecting proper self-supervised pre-text tasks, where the task targets are automatically generated without human labeling. Researchers have tried various pre-text tasks, including context prediction [10], colorization of grayscale images [34], solving jigsaw puzzles [23], the split-brain approach [35], learning to count objects [24], rotation prediction [14], learning to cluster [3], and predicting missing parts [18]. Recently, the attention of this field has mainly shifted to a specific pre-text task of instance discrimination by treating every image as a distinct class [12, 32, 36, 15, 22, 5], which demonstrates performance superior to other pre-text tasks. The PIC framework also follows this line by utilizing instance discrimination as its pre-text task.

**Non-parametric instance discrimination approaches**  The state-of-the-art instance discrimination approaches, i.e. SimCLR [5] and MoCo v2 [15, 6] all follow a dual-branch structure in training, where two augmentation views of each image are required to be sampled at an optimization iteration. The instance discrimination is achieved by encouraging agreement of the two views from the same image and dispersing that of augmentation views from different images.

In general, for dual-branch approaches, special designs are usually required to address the information leakage issue: "*positive and negative pairs can be easily discriminated by leaked information unrelated to feature representation power*". MoCo [15, 6] adopts a shuffle BN and a momentum key encoder. SimCLR [5] adopts a global BN with BN statistics synchronized among GPUs, and employs the samples only in the current batch as negative instances. These special designs increase the framework complexity and sometimes limit the transfer performance. For example, more negative pairs can help with transfer accuracy [15], but [5] uses only negative pairs within the same batch and the number of negative pairs is thus limited.

In contrast to these dual-branch approaches, the PIC framework is based on one branch and naturally will not encounter these information leakage issues.

**Parametric instance discrimination approaches**  The pioneering work on learning visual features by instance discrimination is based on *parametric* instance classification [12]. Though direct in learning the instance discrimination pretext task and simple in use, it performed significantly worse than non-parametric approaches, and was believed to be "*not generalized to new classes or new instances*" [32]. Our framework is basically a revisit of this early approach [12]. We show that the poor performance is not due to its intrinsic limitations, but mainly due to improper component settings. We also propose two novel techniques to address the optimization issue caused by too infrequent visiting of each instance class and to improve practicality regarding training time and GPU memory consumption. With these proper components and the two novel techniques, the PIC framework can now be as effective as or better than the state-of-the-art dual-branch non-parametric approaches [5, 6].

# 4  Experiments

## 4.1  Experimental Settings

We perform unsupervised feature pre-training on the most widely-used dataset, ImageNet-1K [8], which have $\sim$1.28 million training images. For ImageNet-1K, we vary the training lengths from 200 epochs to 1600 epochs[2] to facilitate comparison with previous reported results. In all experiments, a ResNet-50 [16, 17] model is adopted as the backbone network. Eight GPUs of Titan V100 and a total batch size of 512 are adopted. We follow the similar augmentations and training settings as [5, 6], with details shown in Appendix C. For the cosine soft-max loss (1), we find out that $\tau = 0.2$ could generally perform well thus we adopt it for all experiments. For the experiments with recent negative instance sampling, we adopt number of negative instances as $K = 65536$ by default. For sliding window data scheduler, we adopt window size $W = 131072$ and stride $S = 16384$ by default.

To evaluate the quality of pretrained features, we follow common practice [18, 15, 5] to use two types of transfer tasks. The first is a linear classification protocol with *frozen* pre-trained features evaluated

**Table 1:** Applying common **component settings** from other frameworks into PIC.

| strong aug | two-layer FC head | cosine soft-max | ImageNet Top-1 | ImageNet Top-5 |
|:---:|:---:|:---:|:---:|:---:|
| ✓ | ✓ | | 46.5 | 68.7 |
| ✓ | | ✓ | 60.5 | 82.6 |
| | ✓ | ✓ | 62.3 | 84.0 |
| ✓ | ✓ | ✓ | 66.2 | 87.0 |

**Table 2:** Ablation study on **negative instance sampling and classification weight correction**.

| # neg instance | $2^9$ | $2^{10}$ | $2^{12}$ | $2^{14}$ | $2^{16}$ | $2^{18}$ | full |
|:---|:---:|:---:|:---:|:---:|:---:|:---:|:---:|
| w/o correction | 56.8 | 57.6 | 61.0 | 61.6 | 62.3 | 65.6 | 66.2 |
| w. correction | 65.5 | 65.8 | 66.0 | 66.1 | 66.2 | 66.2 | 66.2 |

**Table 3:** Ablation study on the **hyper-parameters of sliding window scheduler**. $^*$ denotes the default setting.

| Window Size $W$ | Stride $S$ | ImageNet Top-1 | ImageNet Top-5 |
|:---:|:---:|:---:|:---:|
| 262144 | 16384 | 67.3 | 87.4 |
| 131072$^*$ | 16384$^*$ | **67.3** | **87.6** |
| 65536 | 16384 | 66.8 | 87.5 |
| 32768 | 16384 | 66.2 | 87.0 |
| 131072 | 32768 | 66.9 | 87.5 |
| 131072$^*$ | 16384$^*$ | **67.3** | **87.6** |
| 131072 | 8192 | 67.1 | 87.5 |
| 131072 | 4096 | 66.5 | 87.2 |

**Table 4:** Ablation study on **sliding window** w.r.t. different training epochs on ImageNet.

| #epoch | Sliding window | ImageNet Top-1 | ImageNet Top-5 |
|:---:|:---:|:---:|:---:|
| 50 | | 53.3 | 76.2 |
| 50 | ✓ | 60.4 | 82.5 |
| 100 | | 62.7 | 84.4 |
| 100 | ✓ | 64.8 | 85.8 |
| 200 | | 66.2 | 87.0 |
| 200 | ✓ | 67.3 | 87.6 |
| 400 | | 68.5 | 88.7 |
| 400 | ✓ | 69.0 | 88.8 |
| 1600 | | 70.4 | 89.8 |
| 1600 | ✓ | 70.8 | 90.0 |

**Table 5:** Comparison of **# augmentations per iteration $\times$ # epochs**.

| Method | #aug/iter$\times$#ep | ImageNet Top-1 | ImageNet Top-5 |
|:---|:---:|:---:|:---:|
| SimCLR [5] | 2$\times$100 | 64.7 | 86.0 |
| MoCo v2 [6] | 2$\times$100 | 64.1 | 85.7 |
| PIC (ours) | 2$\times$100 | 65.0 | 86.2 |
| PIC (ours) | 1$\times$200 | **67.3** | **87.6** |
| SimCLR [5] | 2$\times$200 | 66.6 | 87.3 |
| MoCo v2 [6] | 2$\times$200 | 67.5 | 88.0 |
| PIC (ours) | 2$\times$200 | 67.6 | 88.1 |
| PIC (ours) | 1$\times$400 | **69.0** | **88.8** |

using the same dataset as in the pre-training but with category annotations. The second is to fine-tune pretrained networks to various downstream vision tasks, including semi-supervised ImageNet-1K classification [8], iNaturalist18 fine-grained classification [28], Pascal VOC object detection [13] and Cityscapes semantic segmentation [7]. Please see Appendix C for more experiment details.

## 4.2 Ablation Study

The linear evaluation protocol [18, 15, 5] on the ImageNet-1k dataset is used in ablations.

**Ablation I: component settings from other frameworks** Table 1 shows the performance of 200-epoch pretrained features by PIC using combinations of several recent component settings, including stronger augmentations, two-layer FC head, and cosine soft-max loss. All the three techniques are significantly beneficial, improving PIC to achieve a competitive 66.2% top-1 accuracy on ImageNet-1K linear evaluation. Note that the cosine soft-max loss is especially crucial, incurring about 20% improvement on the top-1 accuracy, which is perhaps the *X factor* bringing the direct classification approaches [12] back to relevance.

**Ablation II: negative instance sampling and classification weight correction** Table 2 ablates the effects of negative instance sampling and classification weight correction on PIC using 200-epoch pre-training. It shows that using $2^{16} = 65536$ negative instances is able to achieve the same accuracy as that without sampling (see the "full" column). The classification weight correction approach is also crucial as the method without it incurs significant performance degradation especially when the number of sampled instances is small.

The GPU memory and actual time are reduced from 8690MiB, 558s/epoch to 5460MiB, 538s/epoch, respectively, by using the default setting of $K = 65536$ negative instances. With negative instance sampling and weight correction approach, the GPU memory consumption and speed is near constant to different data size, making the PIC framework much more practical.

**Ablation III: sliding window data scheduler** Table 3 ablates the two hyper-parameters used in the sliding window scheduler of PIC framework by 200-epoch pre-training, where a window size of 131072 and a stride of 16384 show the best tradeoff of visiting length ($D = 131072$) and majority

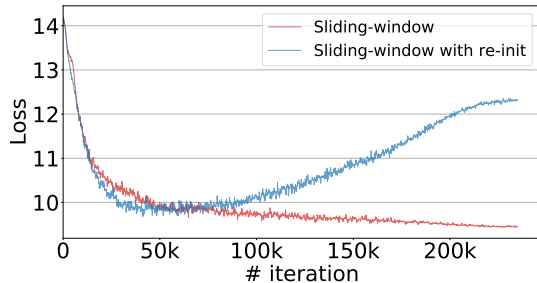

**Figure 3:** This figure illustrates the loss curves of the original sliding-window based training, and the sliding-window based training with re-initializing all the instance class weights which have not been seen for a long time, to understand the opposite effect of not visiting instances for a long time in sliding-window scheduler.

**Table 6:** System-level comparison of linear evaluation protocol with ResNet-50 on ImageNet.

| Method | Linear Eval. | |
|---|---|---|
| | Top-1 | Top-5 |
| Exemplar [12] | 48.6 | - |
| BigBiGAN [11] | 56.0 | - |
| InstDisc. [32] | 54.0 | - |
| Local Agg. [36] | 60.2 | - |
| PIRL [22] | 63.2 | - |
| CPCv2 [18] | 63.8 | 85.3 |
| CMC [27] | 64.1 | - |
| SimCLR [5] | 69.3 | 89.0 |
| MoCo v2 [6] | **71.1** | - |
| PIC (ours) | **70.8** | **90.0** |

**Table 7:** System-level comparison of semi-supervised classification with ResNet-50 on ImageNet.

| Method | Semi-supervised Learning | |
|---|---|---|
| | 1% Labels | 10% Labels |
| InstDisc. [32] | 39.2 | 77.4 |
| PIRL [22] | 57.2 | 83.8 |
| SimCLR [5] | 75.5 | 87.8 |
| PIC (ours) | **77.1** | **88.7** |

**Table 8:** Comparison on transfer learning with ResNet-50.

| Method | iNaturalist 18 | | Pascal VOC | | | Cityscapes |
|---|---|---|---|---|---|---|
| | Top-1 | Top-5 | AP | $AP_{50}$ | $AP_{75}$ | Val mIoU |
| Scratch | 65.4 | 85.5 | 33.8 | 60.2 | 33.1 | 72.7 |
| Supervised | 66.0 | 85.6 | 53.5 | 81.3 | 58.8 | 75.7 |
| MoCo [15] | 65.7 | 85.7 | 57.0 | 82.4 | 63.6 | 76.4 |
| PIC (ours) | 66.2 | 85.7 | 57.1 | 82.4 | 63.4 | 76.6 |

ratio ($\gamma = 87.5\%$). It achieves 67.3% top-1 accuracy on the ImageNet-1K dataset, outperforming the epoch-based scheduler by 1.1%. Also note that all the hyper-parameters in the table lead to higher performance than the epoch-based scheduler, showing a large range of applicable hyper-parameters (the visiting distance $D \in [2^{15}, 2^{18}]$ and the majority ratio $\gamma \in [50\%, 96.88\%]$).

Table 4 shows the results with varying training lengths from 50 epochs to 1600 epochs on ImageNet-1K. By 50-epoch pre-training, the sliding window based scheduler significantly outperforms the previous epoch-based scheduler by 7.1% (60.4% vs 53.3%), indicating that the proposed scheduler significantly benefits optimization. With longer training, its gains over the epoch-based scheduler are smaller, and converge to 0.4% when 1600 epochs are used. We also perform experiments on larger data using ImageNet-11K, shown in Appendix E, and the effectiveness of the sliding window scheduler is also verified.

In the sliding-window scheduler, an instance is repeatedly visited very frequently in a short time, and then not visited for a long time. The performance improvement on sliding-window scheduler should come from repeatedly visiting instances in a short time. To further understand the opposite effect of not visiting instances for a long time in sliding-window scheduler, we perform an experiment by re-initializing all the instance class weights which have not been seen for a long time. Figure 3 illustrates its loss curve compared to that of the original sliding-window based training. At early steps, explicitly forgetting the instances performs similarly well with the original sliding-window based method, indicating the forgetting issue does not affect optimization. This is probably because the large learning rate enables learning classification weights well even when given random initialization for the long-period-not-visited instances. At later steps, the standard sliding-window has steadily reduced loss while explicitly forgetting the instances results in much poorer performance. This indicates that the weights for long-period-not-visited instances may still effect well, although their weights are not updated for a long period, probably because of the small learning rates.

### 4.3 Comparison with other frameworks

**Linear evaluation protocol** We first compare the proposed PIC framework to previous state-of-the-art unsupervised pre-training methods, i.e. MoCo and SimCLR, under similar training lengths such that different methods exploit the same number of total augmentation views, as shown in Table 5. The PIC framework outperforms SimCLR and MoCo v2 by 2.6% and 3.2% respectively on top-1

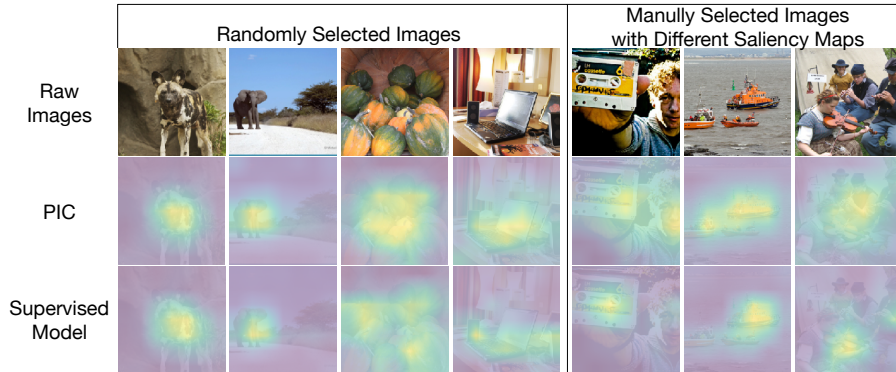

**Figure 4:** Visualizations of the normalized saliency map for randomly selected images (left), and manully selected images with different saliency maps (right).

accuracy when 200-epoch augmentation views are used. The accuracy gains are 2.4% and 1.5% respectively for 400-epoch augmentation views.

The gains of PIC over SimCLR and MoCo v2 partly come from the one-branch nature of the PIC framework, where we observe substantially better results by using $1 \times 200$ and $1 \times 400$ augmentation views over the $2 \times 100$ and $2 \times 200$ settings of two augmentation views per iteration. While SimCLR and MoCo v2 can only perform in a two-branch fashion, the proposed PIC framework can work well using one-branch training, which improves the convergence speed.

We then compare PIC with the previous state-of-the-art approaches, following the recent practice [5, 6] in which longer training lengths are employed, i.e. 1600 epochs (equivalent to 800-epoch training of the two-branch approaches). The proposed PIC framework achieves 70.8% top-1 accuracy on the ImageNet-1K dataset, noticeably outperforming SimCLR (+1.6%) and on par to MoCo v2 (-0.3%). Due to the high similarity with supervised image classification frameworks, techniques from supervised image classification may also be potentially beneficial for the PIC framework, for example, through architectural improvements [25, 20, 33] (PIC can generally use any of them without facing information leakage issues), model ensemble [19] (MoCo may have benefited from it through the momentum key encoder), and large margin loss [30, 9]. We leave them as future works.

**Transfer to downstream tasks** Table 8 and 7 compare the PIC framework with the previous state-of-the-art approaches [22, 5] on several downstream tasks. The PIC framework performs on par or slightly better than previous approaches on these tasks. In particular, PIC achieves the state-of-the-art accuracy on the semi-supervised ImageNet-1K classification task with both 1% and 10% of the labels, outperforming the second best methods by 1.6% and 0.9%, respectively.

## 5 Understanding PIC by Comparing with Supervised Classification

The unsupervised parametric instance classification (PIC) framework and the supervised classification framework both formulate the class discrimination problem by parametric classification. Such similarity in formulation encourages us to study the connections of these two frameworks, so as to further understand the PIC framework.

We first visualize the normalized saliency maps of CNN activations (conv5) to show which region the model tends to focus for discrimination. We compute the L2 Norm for the activations of each spatial location, then normalize over all locations via dividing the sum of them. The visualizations of randomly selected images are shown in the left part of Figure 4, with more examples shown in Appendix G. We can observe that, in most cases, the saliency maps generated by PIC are similar to the supervised pre-trained model. We further measured the similarity between the saliency maps generated by PIC and supervised pre-trained model, which is defined as the sum over minimal values for each location of two saliency maps. The distribution of similarities between the two saliency maps generated by PIC (67.3% top-1 accuracy on linear evaluation) and supervised pre-trained model (77.1% top-1 accuracy) is shown in Figure 5(a), where most pairs have similarity larger than 0.6 and the average is 0.762. For reference, the average similarity between a random saliency map and the

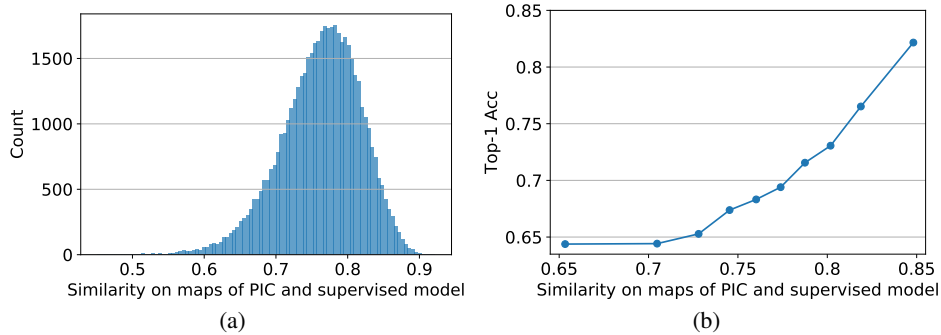

(a)                                                              (b)

**Figure 5:** (a) Statistical results on the counts of images w.r.t different values of similarity on different saliency maps of PIC and supervised model on the ImageNet validation set; (b) Top-1 accuracy w.r.t. similarity of saliency map on the ImageNet validation set.

saliency map of supervised pre-trained model is $0.140$. This clearly shows that the saliency maps of the two models are very similar statistically.

Besides, we manually select several images whose saliency maps of PIC and supervised pre-trained model are different, shown in the right part of Figure 4. We can observe that when there are multiple objects in the images, the saliency map of PIC tends to be distributed on multiple objects. In contrast, that of the supervised pre-trained model tends to focus on a certain single object, which belongs to the specific annotated label. Intuitively, as PIC tends to focus on the most salient area in images to discriminate different images, it may not be biased towards any specific object but choose to distract the attention to all objects. In contrast, the supervised pre-trained model can utilize label information, so it tends to focus on the objects in the label set which could help determine the label.

Furthermore, we evaluate the relationships between the top-1 accuracy of the linear evaluation and the similarity of the saliency maps generated by PIC and supervised pre-trained model. The results are shown in Figure 5(b). We could observe a strong positive correlation between the similarity and top-1 accuracy. If there are any techniques which could make the the saliency maps generated by PIC and supervised pre-trained model to be more similar, the performance may be further improved.

## 6   Conclusion

In this paper, we present a simple and effective framework, parametric instance classification (PIC), for unsupervised feature learning. We show there are no intrinsic limitations within the framework as there are in previous works. By employing several component settings used in other state-of-the-art frameworks, a novel sliding window scheduler to address the extreme infrequent instance visiting issue, and a negative sampling and weight update correction approach to reduce training time and GPU memory consumption, the proposed PIC framework is demonstrated to perform as effectively as the state-of-the-art approaches and shows the practicality to be applied to almost unlimited training images. We hope that the PIC framework will serve as a simple baseline to facilitate future study.

## Broader Impact

Since this work is about unsupervised pre-training, which could be directly adopted in the downstream tasks. Researchers and engineers engaged in visual recognition, object detection and segmentation tasks may benefit from this work. In the future, the people who are engaged in annotating images may be put at disadvantage from this research. If there is any failure in this system, the random initialized model is the lower bound of this unsupervised pre-trained model. This pre-trained model may leverage biases in the dataset used for pre-training, but the biases of unsupervised pre-trained model may be smaller than that of supervised pre-trained model which also used manual annotations.

## Footnotes

*Equal Contribution. The work is done when Zhenda Xie, Bin Liu and Yutong Lin are interns at Microsoft Research Asia.

[2]For the sliding window data scheduler where data is not fed epoch by epoch, we also use the term "epoch" to indicate the training length with iteration numbers equivalent to the epoch-based data scheduler.

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
