[Supplementary Material]

**Algorithm 1:** Pseudocode of PIC in a PyTorch-like style.

```
# f: backbone network tailed with head
# N: number of instances
# D: feature dimension
# W: weight matrix of classifier (D x N)
# t: temperature
# K: #instances sampled by recent sampling

for x, y in sliding_window_data_scheduler:
    # x: a mini-batch with B samples
    # y: instance indices

    W1 = sample(W) # recent sampling
    W1 = correct(W1) # weight correction

    x = aug(x) # randomly augment

    feat = f.forward(x) # feature: B x D

    # logits: B x K
    logits = cos_sim(feat, W1)

    # cross entropy loss
    loss = CrossEntropyLoss(logits/t, y)

    # SGD update
    loss.backward()
    update(f.params)
    update(W1.params)
```

**Algorithm 2:** Pseudocode of supervised image classification in a PyTorch-like style.

```
# f: backbone network tailed with head
# C: number of classes
# D: feature dimension
# W: weight matrix of classifier (D x C)

for x, y in epoch_based_data_scheduler:
    # x: a minibatch with B samples
    # y: image labels

    x = aug(x) # randomly augment

    feat = f.forward(x) # feature: B x D

    # logits: B x C
    logits = mm(feat, W)

    # cross entropy loss
    loss = CrossEntropyLoss(logits, y)

    # SGD update
    loss.backward()
    update(f.params)
    update(W.params)
```

**Algorithm 3:** Implementation of sliding window data scheduler in PyTorch.

```
import numpy as np
from torch.utils.data import DataLoader, Sampler

class SlidingWindowSampler(Sampler):
    def __init__(self, indices, window_stride, window_size):
        self.indices = indices
        self.window_stride = window_stride
        self.window_size = window_size
        self.start_index = 0
        self.dataset_size = len(self.indices)

        np.random.shuffle(self.indices)

    def __iter__(self):
        # get indices of sampler in the current window
        indices = np.mod(np.arange(self.window_size) + self.start_index, self.dataset_size)
        window_indices = self.indices[indices]

        # shuffle the current window
        np.random.shuffle(window_indices)

        # move start index to next window
        self.start_index = (self.start_index + self.window_stride) % self.dataset_size

        return iter(window_indices.tolist())

dataset = ...
sampler = SlidingWindowSampler(np.arange(len(dataset)), WINDOW_STRIDE, WINDOW_SIZE)
loader = DataLoader(dataset, sampler=sampler, ...)
```

# A   Algorithm

Algorithm 1 and 2 show the pseudocode of PIC and traditional supervised classification in a PyTorch-like style, respectively, which show that PIC can be easily adapted from supervised classification by only modifying a few lines of code.

Furthermore, the simplified PyTorch implementation of sliding window data scheduler is shown in Algorithm 3.

## B  Detailed deduction and implementation for weight update correction on weight decay and momentum terms

Considering the stochastic gradient descent combined with momentum and weight decay, the parameters will be updated as follow:

$$\mathbf{u}_i^{(t+1)} := m\mathbf{u}_i^{(t)} + (\mathbf{g}_i^{(t)} + \lambda\mathbf{w}_i^{(t)}), \quad \mathbf{w}_i^{(t+1)} := \mathbf{w}_i^{(t)} - \eta\mathbf{u}_i^{(t+1)}, \tag{4}$$

where $\mathbf{g}_i^{(t)}$ and $\mathbf{u}_i^{(t)}$ are the gradient and momentum vectors at iteration $t$ for classification weight $\mathbf{w}_i^{(t)}$ of class $i$, respectively; $\lambda$, $m$ and $\eta$ are the weight decay, momentum scalar and learning rate, respectively.

When we adopt the *recent sampling* strategy, those instance examples not included in the recent iterations will have zero gradient during training. Therefore, we can replace $\mathbf{g}_i^{(t)}$ with $\mathbf{0}$ in the above equation and simplify it by using the matrix format as follow:

$$\begin{pmatrix} \mathbf{w}_i^{(t+1)} \\ \mathbf{u}_i^{(t+1)} \end{pmatrix} := \begin{pmatrix} 1 & -\eta \\ 0 & 1 \end{pmatrix} \begin{pmatrix} 1 & 0 \\ \lambda & m \end{pmatrix} \begin{pmatrix} \mathbf{w}_i^{(t)} \\ \mathbf{u}_i^{(t)} \end{pmatrix}. \tag{5}$$

Then, we can merge these two simple matrices and treat them as a transfer matrix, and we can easily calculate the $t'$-step update, which has been demonstrated in Section 2.3.

$$\begin{pmatrix} \mathbf{w}_i^{(t+t')} \\ \mathbf{u}_i^{(t+t')} \end{pmatrix} := \begin{pmatrix} 1 - \eta \cdot \lambda & -\eta \cdot m \\ \lambda & m \end{pmatrix}^{t'} \begin{pmatrix} \mathbf{w}_i^{(t)} \\ \mathbf{u}_i^{(t)} \end{pmatrix}. \tag{6}$$

## C  Experimental Settings

**Pre-training**  We follow the similar augmentation as Chen et al. [5] to adopt random resize and crop, random flip, strong color distortions, and Gaussian blur as the data augmentations, where the only difference is that we adopt the crop scale as 0.2 as Chen et al. [6]. We use Stochastic Gradient Descent (SGD) as our optimizer, with weight decay of 0.0001 and momentum as 0.9. We adopt a batch size of 512 in 8 GPUs with batch size per GPU as 64. The learning rate is initialized as 0.06 and decayed in cosine schedule. For different training epochs, we all adopt a 5-epoch linear warm-up period. Different to MoCo [15] and SimCLR [5] which use ShuffleBN or SyncBN tricks to alleviate the information leakage issue, we only need the standard batch normalization, which computes the statistics inside each GPU. For the cosine soft-max loss (1), we find out that $\tau = 0.2$ could generally perform well thus we adopt it for all experiments. For the experiments with recent negative instance sampling, we adopt number of negative instances as $K = 2^{16} = 65536$ by default. For sliding window data scheduler, we adopt window size $W = 2^{17} = 131072$ and stride $S = 2^{14} = 16384$ by default.

**Linear Evaluation Protocol**  In linear evaluation, we follow the common setting [6, 5] to freeze the backbone of ResNet-50 and train a supervised linear classifier on the global average pooling features for 100 epochs. Note that, the 2-layer head in unsupervised pre-training is not used in the linear evaluation stage. We adopt a batch size of 256 in 8 GPUs, SGD optimizer with momentum 0.9, initial learning rate of 30 with a cosine learning rate scheduler, weight decay of 0, and data augmentations composed by random resize and crop with random flip. The standard metrics of top-1 and top-5 classification accuracy with one center crop are reported on the ImageNet validation set.

**Semi-supervised classification on ImageNet**  In semi-supervised learning, we follow the standard setting [5] to fine-tune the whole backbone of ResNet-50 and train a linear classifier on top of the backbone from scratch. The data augmentations are random resize and crop with random flip. We

**Table 9:** Top-1 and Top-5 linear classification accuracy of 5 trials.

| Sliding Window | Trials | 1 | 2 | 3 | 4 | 5 | overall |
|---|---|---|---|---|---|---|---|
| | Top-1 | 66.4 | 66.3 | 66.2 | 66.2 | 66.1 | 66.24±0.11 |
| | Top-5 | 87.0 | 87.1 | 87.1 | 86.9 | 87.0 | 87.02±0.08 |
| ✓ | Top-1 | 67.4 | 67.4 | 67.3 | 67.3 | 67.2 | 67.32±0.08 |
| ✓ | Top-5 | 87.6 | 87.7 | 87.6 | 87.6 | 87.5 | 87.60±0.07 |

optimize the model with SGD, using a batch size of 256, a momentum of 0.9 and a weight decay of 0.00003. We train the model for 10 epochs for 1% labels and 20 epochs for 10% labels. In addition, we use the cosine learning rate scheduler without the warm-up stage.

**Fine-grained Image Classification on iNaturalist 2018** In fine-grained image classification on iNaturalist 2018, we fine-tune the whole backbone of ResNet-50 and learn a new linear classifier on top of the backbone for 100 epochs. Similar to the pre-training stage, we adopt a cosine classifier with temperature parameter by default. We optimize the model with SGD, using a batch size of 256, a momentum of 0.9, a weight decay of 0.0001 and a $\tau$ of 0.03. The initial learning rate is 0.2 and a cosine learning rate scheduler is adopted. The standard metrics of top-1 and top-5 classification accuracy with center crop are reported on iNaturalist 2018 validation set.

**Object Detection on Pascal VOC** In object detection on Pascal VOC, the detector is Faster R-CNN [26] with a backbone of ResNet50-C4, which means the backbone ends with the $conv_4$ stage, while the box prediction head consists of $conv_5$ stage (including global pooling) followed by a BatchNorm layer. We use Detectron2 [31] implementation to fine-tune all layers in an end-to-end manner. During training, the image scale is $[480, 800]$ while the image scale is 800 at the inference stage. We optimize the model with SGD for 24k iterations on Pascal VOC `trainval07+12` set, using a batch size of 16, a momentum of 0.9 and a weight decay of 0.0001. The learning rate is 0.02 (multiplied by 0.1 at 18k and 22k iterations). The standard metrics of AP50, AP75 and mAP are reported on Pascal VOC `test2007` set.

**Semantic Segmentation on Cityscapes** In semantic segmentation on Cityscapes, we follow [15] and use a similar FCN-16s [21] structure. Specifically, all convolutional layers in the backbone are copied from the pre-trained ResNet50 model, and the $3 \times 3$ convolutions in $conv_5$ stage have dilation 2 and stride 1. This is followed by two extra $3 \times 3$ convolutions of 256 channels with BN and ReLU, and then a $1 \times 1$ convolution for per pixel classification. Also, we follow the same large field-of-view design in [4] similar to [15] and set dilation = 6 in those two extra $3 \times 3$ convolutions.

During training, we augment the image with random scaling from 0.5 to 2.0, crop size of 769 and random flip. The whole image of single scale is adopted in inference stage. We optimize the model with SGD for 60k iterations on the Cityscapes `train_fine` set, using a batch size of 8, a momentum of 0.9 and a weight decay of 0.0001. The initial learning rate is 0.01, a cosine learning rate scheduler and synchronous batch normalization are adopted. The standard metric of mean IoU is reported on the Cityscapes `val` set.

# D    Experimental results on multiple trials

Here we run 5 trials of PIC pre-training and perform linear evaluation to measure the stability of the evaluation metrics of top-1 and top-5 accuracy. The top-1 and top-5 accuracy results are reported in Table 9. We could observe that the performance of PIC framework is as stable as MoCo v2 with standard deviation of 0.1[3].

**Table 10:** Ablation on **sliding window** on ImageNet-11K dataset.

| Epochs | 25 | 50 | 100 |
|---|---|---|---|
| supervised | - | - | 38.10 |
| PIC w/o sliding window | 22.38 | 31.03 | 32.02 |
| PIC w. sliding window | 30.73 | 31.97 | 32.94 |

**Figure 6:** (a) Linear Evaluation on ImageNet; (b) Object Detection on Pascal VOC.

# E  Experimental Results on ImageNet-11K

Here we perform experiments on a larger-scale image dataset, ImageNet-11K [8], which contains $\sim$7.47 million training images[4]. We follow the same pre-training setting as in ImageNet-1K with only dataset changed, and perform linear evaluation also on ImageNet-11K dataset. Table 10 shows the top-1 accuracy results with varying training lengths from 25 epochs to 100 epochs on ImageNet-11K. By 25-epoch pre-training, the sliding window based scheduler significantly outperforms the previous epoch-based scheduler by 8.35% (30.73% vs 22.38%), indicating that the proposed scheduler significantly benefits optimization. With longer training, its gains over the epoch-based scheduler are smaller, and converge to 0.92% when 100 epochs are used. Also, for 100-epoch pre-training, the gap between PIC and supervised model is 7.16%, which is not large.

For ImageNet-11K dataset, we observed "out-of-memory" issue using full negative instances on the V100 GPU with 16GB GPU memory. Actually, the GPU memory consuming is 17.26G. By using the default setting of sampling $K = 65536$ negative instances with weight correction, the GPU memory and actual speed remain unchanged as ImageNet-1K of 5460MiB and 3,140s/epoch.

# F  Compared to Supervised Classification with Varying Classes

From the perspective of optimization goals, the only difference between the parametric instance classification framework and supervised classification framework is how to define the classes for each instance. Taking ImageNet-1K dataset as an example, images are divided into 1k groups in supervised classification. In contrast, each image in the PIC framework is considered as a single category. Considering this relationship, could we find a better partition with the number of classes larger than 1K and less than the number of instances?

To this end, we study how the performance changes under different kinds of category partitions. In order to smoothly transform under the two category partitions of PIC and supervised classification, we proposed a constrained category construction scheme based on clustering: images belonging to the same category in supervised classification are divide into the sub-categories. In this way, images that do not belong to the same category in supervised classification will not be divided into the same category under the new category partition in any case.

We trained a series of models for different numbers of categories (1K, 2K, 4K, 16k, 64K and 128K). We use a variant of K-means [2] as our clustering method to ensure that each cluster has the same number of instances. And we use the features generated by supervised pre-trained model and euclidean distance as the distance measure in the clustering to generate new categories. It is worth noting that the setting of 1K categories is the supervised classification and the setting of 128K is the PIC framework.

A series of models are trained for each new category partition with the same setting as PIC using 200 epochs, and evaluated on ImageNet (linear evaluation) and Pascal VOC (object detection). The results are shown in Figure. 6. There are two observations:

- The larger the number of classes, the lower the linear evaluation accuracy. This is reasonable because that the linear evaluation is performed on the supervised classification of 1k classes.

- Models with higher accuracy of linear evaluation on ImageNet do not perform significantly better on Pascal VOC, which indicates the linear evaluation performance is not a good indicator to evaluate transferability of the pre-trained models.

## G    More visualizations of the normalized saliency maps

Here we show more visualizations of the normalized saliency maps generated by our PIC model and the supervised training model in Figure 7. The first six rows with randomly selected images show that the saliency maps generated by PIC and the supervised pre-trained model are similar in the most cases. The last three rows contain manually selected images whose saliency maps of PIC and supervised pre-trained model are different. We can have the similar observation as claimed in the main paper that when there are multiple objects in the images, the saliency map of PIC tends to be distributed on multiple objects. In contrast, that of the supervised pre-trained model tends to focus on a certain single object, which belongs to the specific annotated label.

**Figure 7:** More visualizations of the normalized saliency map.

## Footnotes

[3]`https://github.com/facebookresearch/moco`

[4]The original ImageNet-11K datasets have about 10.76M total images. We download all images that are still valid, constituting 7.47M total images.