[Reviews · NeurIPS 2020]

Review 1

Summary and Contributions: This paper presents a novel parametric instance classification (PIC) method to perform one-branch parametric instance classification. Additionally, to address infrequent instance visiting and training consumption, they introduce two novel techniques, sliding window data scheduler and negative instance class sampling along with weight update correction.

Strengths: Novelty: This paper has the solid contribution, its methods are reasonable and show solid improvements. The intuition of the proposed methods is easy to understand. The method may serve as a simple baseline in future. Experiments: The ablation study is quite extensive and supports the effectiveness of the proposed methods. Writing: The writing is good and easy to read. Figures are illustrative and helpful for the readers to understand the proposed methods. Equations are clear and with enough explanation.

Weaknesses: There are many implementation details in the proposed methods, so it is better to release the codes as soon as possible.

Correctness: The claims and method in this paper are technically correct, which is also supported with extensive experiments.

Clarity: The paper is well prepared and written.

Relation to Prior Work: The difference with previous works is clearly claimed.

Reproducibility: Yes

Additional Feedback: After reading the rebuttal, the authors address all my concerns. I maintain my score.


Review 2

Summary and Contributions: The paper proposes an updated implementation of instance discrimination for self-supervised learning called parametric instance classification (PIC). The main contribution is that by using implementation tricks from recent unsupervised frameworks PIC obtains similar performance to recent approaches like SimCLR while being a simpler model.

Strengths: The paper shows a good analysis of the implementation components of PIC. The ablation studies are thorough and show that each part of the implementation needs to be tuned to achieve good performance. I think there is value on showing the simple approaches for self-supervised learning like PIC can attain similar performance to more complex approaches like SimCLR.

Weaknesses: The main weakness of this paper is novelty. Most of the implementation tricks used on this paper are introduced in other papers (as the authors correctly point out). The novel contributions are in data scheduler and in the negative sample strategy. Which in my opinion do not constitute a contribution worth publication at NeurIPS. Another weakness of this paper is on the empirical side. Especially on the comparison with SimCLR. As a reviewer, I understand that computational resources play a big role on this comparison. Since we cannot expect all institutions to have access to the same compute resources. Tab 5 is the main example of this situation. The proposed approach seems to outperform SimCLR if training for 200 epochs. However, when SimCLR is trained for 1000 epochs it outperforms the proposed method. From this results I cannot say if the proposed approach is outperforming SimCLR, it could be that the proposed approach benefits from the cross-level discrimination earlier in training, and thus outperforms SimCLR at epoch 200, but then SimCLR catches up and outperforms the proposed approach when converged.

Correctness: The submission is empirically and methodologically correct. However, I am concerned that due to computational resources, a fair comparison between the proposed method and SimCLR cannot be obtained.

Clarity: The paper is clear but a lot of notation details are not properly introduced or used. For example: - line 72 W \in R^{D \times N}, what is D and N?

Relation to Prior Work: Related work is properly discussed and how this submission differs from previous contributions is clear. Although there are a couple of missing references on cross-level relationship modeling: Bautista, Miguel A., et al. "Cliquecnn: Deep unsupervised exemplar learning." Advances in Neural Information Processing Systems. 2016. Milbich, Timo, et al. "Unsupervised video understanding by reconciliation of posture similarities." Proceedings of the IEEE International Conference on Computer Vision. 2017. Bautista, Miguel A., Artsiom Sanakoyeu, and Bjorn Ommer. "Deep unsupervised similarity learning using partially ordered sets." Proceedings of the IEEE Conference on Computer Vision and Pattern Recognition. 2017.

Reproducibility: Yes

Additional Feedback:


Review 3

Summary and Contributions: The work tackles the problem of self-supervised representation learning. It shows how to "fix" the parametric approach of [12] such that it performs better or on par with the most recent dual-branch contrastive frameworks while the training procedure is efficient. To this end, the authors propose a sliding-window sampling strategy to alleviate infrequent instance class visits. For improving training efficiency, the paper proposes to sample fewer negative classes and to correct classification weights only when they are used in forward pass, instead of updating them on every iteration.

Strengths: 1. This work addresses an important problem of self-supervised feature learning with a simpler solution than the current SoTA methods while delivering comparable results. The proposed solution pays a lot of attention to small implementation details, and I appreciate that. 2. The experimental section is reach with results and ablation studies. The authors provide enough evidence that their method achieves results better than the current SoTA. 3. The paper is well written and overall clear. 4. The related work is well organized and mentions relevant articles.

Weaknesses: 1. The authors must be more clear in the introduction that the proposed solution is a "fix" of [12], rather than a new PIC approach, as introduced in lines 29-30 by saying: "... This paper presents a framework which solves instance discrimination by direct parametric instance classification (PIC)". This framework has been already proposed by [12] and the authors must mention it. 2. It is not clear to me why exactly the sliding-window data sampler improves training. My understanding is that with the sliding-window sampler, an instance is repeatedly visited several (something like B/S) times in a row, and then not visited for a very long time (something like B * N / S). This means that in the expectation, a single instance class is visited as often as it would have been visited with epoch-based training. Does this mean that the improvement in training comes only from being able to "learn well" a single instance class, before moving to another one? How about the opopsit effect like forgetting this instance class [1*], since the network does not see this instance class for a much longer period after it has been repeatedly visited? The paper is lacking a clear explanation of this phenomena and hence the sliding window sampling is not well motivated. 3. While it is nice to have Section 5, the feature visualization technique used there is not limited to models with parametric classifiers. Therefore, it would be much more valuable if we could see a comparison of visualizations and statistics (Figure 3) with other methods such as MoCo and SimCLR. Otherwise, simply stating the facts only for PIC without any comparisons is not very informative. [1*] - Toneva et. al "AN EMPIRICAL STUDY OF EXAMPLE FORGETTING DURING DEEP NEURAL NETWORK LEARNING"

Correctness: The claims in the paper seem to be correct.

Clarity: The paper is clear and well written.

Relation to Prior Work: In general, yes. See my comment about [12] above.

Reproducibility: Yes

Additional Feedback: 1. I do not fully understand how the authors count epochs of training, as for example in line 232, while using sliding window sampler, that does not have a clear notion of epochs. Could the authors please elaborate on that? Overall, I like the work and think that it is an important direction. I am willing to adjust my score if the authors address the comments above. ---------------------------- After Rebuttal --------------------------- The authors addressed my comments and provided more insights into the method which is very valuable. I suggest the authors to include their clarifications in the final version of the paper. I recommend the paper for acceptance.


Review 4

Summary and Contributions: This paper proposes a simple framework for unsupervised feature learning. Instead of using the current popular dual-branch non-parametric instance classification setting, the proposed method adopts a single-branch parametric setting. The proposed sliding-window data scheduler and negative sampling with weight correcting techniques make the whole framework efficient and practical.

Strengths: 1. Experiments - This paper provides a detailed ablation study to validate the effectiveness of the proposed techniques - The proposed method achieves a new state-of-the-art performance on several downstream visual tasks - The authors also analyze the connection between the proposed method and the supervised training method, which I found interesting and I believe can give the community some insights 2. Significance and Novelty The core contribution of this paper lies in the two training techniques (sliding-window data scheduler and negative sampling with weight correcting). Unlike current state-of-the-art unsupervised feature learning methods that require special handling to avoid data leakage, the proposed method can be easily implemented. I think this could be a good contribution to the community.

Weaknesses: 1. Experiments It will be interesting to know if the proposed method can achieve good performance on larger backbone networks (e.g., deeper or wider) or not. 2. Other questions 1) I wonder if the projection head is dropped in the downstream task 2) Can the authors comment on why cosine softmax brings such a significant improvement in the current setting?

Correctness: The extensive experiments well support and validate the claims made by the authors.

Clarity: Yes. Overall, this paper is easy to read and understand.

Relation to Prior Work: Yes. Instead of using a dual-branch non-parametric setting, the proposed method is a revisit of the single-branch parametric setting. The core contribution lies in the two proposed techniques.

Reproducibility: Yes

Additional Feedback: Update: I have read the comments from other reviewers and the rebuttal. I think the rebuttal well addresses most of the raised issues. Hence, I keep my original rating and recommend acceptance.

[Author Response · NeurIPS 2020]

We thank all reviewers for their insightful and constructive comments.

**Response to R#2 Q2.1:...The main weakness is novelty ... the novel contributions are in data scheduler and negative sample strategy, which in my opinion do not constitute a contribution worth publication at NeurIPS...**
While R#2 well noticed our new techniques, our contribution of *bringing a simple and effective line back to the sight for unsupervised learning* may be overlooked. It breaks the inherent belief that parametric instance classification has intrinsic limitation [32]. Such knowledge is new to the community and is better valued independent of tech contribution. As for the two novel techniques, we respect the reviewer's personal opinion, but we would greatly appreciate if the reviewer could also take the following facts into consideration: 1) "a simple baseline" (by R#1, R#3, R#4) which does not "require special handling to avoid data leakage and can be easily implemented" (by R#4); 2) simple and effective solutions to address *crucial* issues in parametric instance classification (by R#1 ,R#3 ,R#4): the sliding window scheduler addresses the extremely infrequent instance visiting issue and the *corrected updating technique* enables applying the PIC framework to unlimited data scale.

**Q2.2: ...Another weakness of this paper is on the empirical side...when SimCLR is trained for 1000 epochs it outperforms the proposed method...** SimCLR performs worse than our approach in longer training settings (69.3% vs 70.8% in Table 6). Nevertheless, the main goal of this work is not to achieve higher accuracy number, but more importantly to recall another line of unsupervised learning, which could increase the diversity of research.

**Q2.3: ...notation details not properly introduced...** We would check it carefully and improve the writing accordingly.

**Q2.4: ...missing references on cross-level relationship modeling...** The listed papers perform unsupervised learning via pairwise similarity learning. We will add discussion of these papers in the revision.

**Response to R#3 Q3.1: ... make it clear in introduction that the proposed solution is a "fix" of [12] ...** Thanks for the suggestion. We will clarify it in the introduction like "our paper is basically a revisit of [12]" (L184-188).

**Q3.2: ...does the improvement in training comes only from being able to "learn well" a single instance class before moving to another one? ...** Yes, the improvement in training should come from being able to "learn well" a single instance class because the opposite effect could not help training.

**Q3.3: How about the opposite effect like forgetting this instance class [1*] ...** To answer this question, we consider an experiment by re-initializing all the instance class weights which have not been seen for a long time (12.5% in each window). The figure illustrates its loss curve compared to that of the original sliding-window based training. At early steps, explicitly forgetting the instances performs similarly well with the original sliding-window based method, indicating the forgetting issue does not affect optimization. This

is probably because the large learning rate enables learning classification weights well even when given random initialization for the long-period-not-visited instances. At later steps, the standard sliding-window has steadily reduced loss while explicitly forgetting the instances results in much poorer performance. This indicates that the weights for long-period-not-visited instances may still effect well, although their weights are not updated for a long period, probably because of the small learning rates. On the whole, the forgetting of the *minority* may not be a serious issue, and the benefit of well learning the *majority* weights overcomes such disadvantages. We will add discussion in the revision.

**Q3.4: ...it would be much more valuable if we could see a comparison of visualizations and statistics (Fig. 3) with other methods such as MoCo and SimCLR...** Thanks for the suggestion. We conducted similar experiments for MoCo and SimCLR and their behavior is similar as PIC. Actually, the goal of this section is not to compare PIC with MoCo/SimCLR, but to understand PIC itself, where the conclusion may be generalized to other instance discrimination based methods. Note we also perform experiments distinct to PIC, shown in Appendix Sec. F and Fig. 4, which build a smooth transition between PIC and supervised method.

**Q3.5: ...how to count epochs for sliding window sampler...** As stated in the footnote of page 5, we use the term "epoch" to indicate the equivalent training length with that of epoch-based scheduler, to simplify the description.

**Response to R#4 Q4.1: ...results on larger backbone (e.g., deeper or wider)...** ResNet-50 ($2\times$) with 200-epoch pre-training achieves 71.2% top-1 accuracy, which is 3.9% better than that of the standard ResNet-50 ($1\times$, 67.3%).

**Q4.2: ...if projection head is dropped...** Yes, we follow [5,15,22] to drop the projection head in downstream tasks.

**Q4.3: ...comment on why cosine softmax brings such a significant improvement in the current setting...** The only difference between cosine softmax loss and standard softmax loss is that the standard softmax loss accounts vector *magnitudes* for similarity computation, in addition to the *angles* between two vectors used by cosine softmax. We think the significant accuracy drop by standard softmax loss is due to the significantly worse generalization ability of vector *magnitudes* than *angles*. In fact, similar behavior has been widely observed in numerous applications of metric learning, including face recognition [9, 30], person re-ID ("Deep Cosine Metric Learning for Person Re-Identification , WACV-18") and few-shot learning ("MatchingNet-NeurIPS16", "Improving Generalization via Scalable Neighborhood Component Analysis, ECCV-18", "A Closer Look at Few-shot Classification, ICLR-20"). To our knowledge, strict proof of such behavior is still an open question.

[Meta-Review · NeurIPS 2020]

This paper addresses self-supervised feature learning, providing a simpler method than previous work, with comparable performance. The reviewers were unanimous in their decision to accept this paper.